# Association between Occupation and Cervical Disc Degeneration in 1211 Asymptomatic Subjects

**DOI:** 10.3390/jcm11123301

**Published:** 2022-06-09

**Authors:** Masaaki Machino, Hiroaki Nakashima, Keigo Ito, Kei Ando, Sadayuki Ito, Fumihiko Kato, Shiro Imagama

**Affiliations:** 1Department of Orthopedic Surgery, Nagoya University Graduate School of Medicine, 65 Tsurumai, Showa-ku, Nagoya 466-8550, Japan; hirospine@med.nagoya-u.ac.jp (H.N.); andokei@med.nagoya-u.ac.jp (K.A.); sadaito@med.nagoya-u.ac.jp (S.I.); imagama@med.nagoya-u.ac.jp (S.I.); 2Department of Orthopedic Surgery, Chubu Rosai Hospital, Japan Organization of Occupational Health and Safety, 1-10-6 Komei, Minato-ku, Nagoya 455-8530, Japan; keiort@aol.com (K.I.); f.kato.ort@chubuh.johas.go.jp (F.K.); 3Chubu Rosai Nursing School, Japan Organization of Occupational Health and Safety, Nagoya, 1-10-6 Komei, Minato-ku, Nagoya 455-8530, Japan

**Keywords:** cervical disc degeneration, magnetic resonance image, asymptomatic subjects, occupation, disc degeneration

## Abstract

Magnetic resonance imaging (MRI) system has frequently observed degenerative changes in the cervical discs of healthy subjects. Although there are concerns regarding the link between an individual’s occupation and intervertebral disc degeneration (IDD) in the cervical spine, whether the occupation affects IDD is still not clear. This study aimed to evaluate the occupation and IDD interplay using cervical spine MRI among a cohort of healthy individuals, and to evaluate any association between the type of labor and IDD. Using MRI, we prospectively measured at every level, the anteroposterior (AP) intervertebral disc diameter and disc height, in a cohort of 1211 healthy volunteers (606 (50%) male; mean age, 49.5 years). Using a minimum of 100 male and female each from the third to eighth decades of age (20–79 years), IDD was evaluated based on the modified Pfirrmann classification system to derive a disc degeneration score (DDS). We also measured the AP diameters of disc protrusion and of the dural sac as well as the spinal cord. The overall DDS and number of disc protrusions increased with age. Among 11 occupations, there were no significant differences in AP diameter of the dural sac as well as the spinal cord. For the four labor types (heavy object handling, same position maintenance, cervical extension position, and cervical flexion position), there were no significant differences in overall DDS and number of disc protrusions, with or without work. Also, among the four labor types, there were no significant differences in the AP diameter of the dural sac as well as the spinal cord. In this cross-sectional survey of cervical spine MRI data among healthy adult volunteers, occupation and type of labor might have no effect on IDD in the cervical spine.

## 1. Introduction

The conventional concept regarding intervertebral disc degeneration (IDD) associates it with normal aging and alterations in relation to lifetime physical loading. However, it is crucial from an epidemiological standpoint to define a case of IDD and its measurement. A standardized case definition for IDD is lacking. For this reason, it is difficult to compare between studies. Clinically, IDD is largely defined by its method of evaluation. For example, disc height and osteophytes can most usefully be assessed by radiography or computed tomography [1,2], whereas structural changes in the disc, like protrusion or bulging, are best assessed by magnetic resonance imaging (MRI) [3,4]. For studies involving large populations, the preferred technique is using MRI. Most assessment systems involve an evaluation of the disc height, signal intensity, bulging, and osteophytes. The burden of these parameters substantially differs among previous studies [5,6,7,8].

Several factors, possibly causative, have been implicated in IDD, for example, older age, genetic mutations and family history, malnutrition, toxicity from any cause, metabolic derangements, occult infection, neurogenic factors, autoimmune disorders, and mechanical factors [9,10,11,12]. Population studies involving cervical pain and disc disease are minimal compared to those involving the lumbar spine [13,14]. Cervical disc disease deserves further study that includes occupational factors.

Although the relationship between occupation and IDD of the cervical spine is important, it remains unclear whether occupation affects IDD. Therefore, we conducted a large-scale study, across sexes and ages, to investigate the relationship between occupation and IDD. We also investigated whether the type of labor affected the degree of IDD, disc protrusion, and diameter of dural sac as well as the spinal cord. This study aimed to evaluate the occupation and IDD interplay using cervical spine MRI among a large cohort of healthy individuals. We also evaluated the relationship between the type of labor and IDD.

## 2. Materials and Methods

### 2.1. Study Population

Cervical spine MRI assessments were carried out among 1230 healthy volunteers who do not have any neurologic problems. We used mass media advertisements and advocacy posters distributed in health facilities to recruit at least 100 males and 100 females of each per decade of life whose ages ranged from 20 to 79 years (Table 1). The volunteers who had a history of brain or spinal surgery, co-existing neurological conditions (e.g., cerebral infarction or neuropathy), clinical evidence of sensory or motor disorders (like numbness, clumsiness, motor weakness, gait disturbance), and severe neck pain were excluded from the study. Pregnant women, patients with claustrophobia or other contraindications to MRI, individuals who received worker’s compensation, and those whose problems occurred following a motor accident, were excluded. These subjects included participants with a variety of comorbidities, such as smoking, diabetes, and hypertension. Each participant gave written informed consent, and the study was approved by the hospital’s ethics and research committee granted.

We excluded 19 subjects because of measurement challenges or the existence of dental implants. MRI and radiographic data for the final analysis were obtained from 1211 subjects (606 males, 605 females; mean age, 49.5 ± 16.8 years). The subjects’ occupations were investigated; they included office worker, doctor, nurse, medical coworker, housekeeper, service provider, builder, teacher, salesperson, manufacturer, student, carrier, farmer, and other. We also examined four types of labor, involving (1) heavy object handling, (2) same position maintenance, (3) a cervical extension position, and (4) a cervical flexion position.

MRI images were obtained using a 1.5-Tesla superconductive magnet (Signa Horizon Excite HD version 12; GE Healthcare, Chalfont Saint Giles, UK) at a slice thickness of 3 mm in the sagittal plane. T2-weighted images (fast spin echo; TR, 3500 ms; TE, 102 ms) were obtained in the sagittal plane. All images were transferred in the Digital Imaging and Communications in Medicine format to a computer using imaging analytics software (Osiris 4; Icestar Media, Colchester, UK). Using a slice in which disc protrusion was most prominent in sagittal images, we measured the anteroposterior (AP) diameter of the protrusion from the standard line to the posterior top of the protrusion. We defined disc protrusion as intervertebral disc protrusion of more than 1 mm posteriorly. We also measured the AP diameters of the dural sac as well as the spinal cord at the C2/3 and C5/6 levels. IDD was defined using the modified Pfirrmann classification system (Figure 1) [4]. Two spinal surgeons performed subjective grading. In order to enable comparative analysis of IDD severity across ages and sexes, a total score of disc degeneration at six levels (from C2/C3 to C7/T1), termed the disc degeneration score (DDS), was estimated by summing the Pfirrmann scores at each level. A DDS score of 6 indicates that degeneration at all six levels is minimal. A maximum score of 24 indicates that all six levels have Grade IV degeneration [15].

### 2.2. Statistical Analysis

All analyses were carried out with SPSS version 25.0 (IBM, Armonk, NY, USA). Continuous variables are expressed as mean ± standard deviation. The Mann–Whitney U-test was used to compare mean differences between two groups. A *p*-value of < 0.05 indicated statistical significance.

## 3. Results

This study included 1211 asymptomatic subjects, with similar distribution across age groups from the third to eighth decade of age (Table 1). About half of the subjects had passive occupations (e.g., office worker, service provider, and teacher). Physically demanding occupations (e.g., housekeeper, builder, manufacturer) were held by 28% of the subjects (Table 2).

In females, the subjects had heavy object handling work, whereas those who were younger than them had no heavy object handling work. Similarly, in females, the subjects had the same position maintenance work, whereas those who were younger than them had no same position maintenance work. Across both genders, there were no significant differences in age between those who performed work in a cervical extension or a cervical flexion position and those who did not (Table 3).

After occupations were listed in ascending order of age, for all occupations, the overall DDS and number of disc protrusions increased with age. Across all occupations, there were no significant differences in the AP diameter of the dural sac as well as the spinal cord at the C2/3 or C5/6 levels (Figure 2).

In males (Figure 3) and in females (Figure 4), there were no significant differences in overall DDS and number of disc protrusions between each of the four labor types, nor any significant differences in the AP diameter of the dural sac as well as the spinal cord at the C2/3 or C5/6 levels.

## 4. Discussion

To our knowledge, this is the first study to use cervical MRI to demonstrate the relationship between occupation and the degree of IDD in asymptomatic subjects. In this large-scale, cross-sectional study of relatively healthy subjects, occupation and type of labor appeared to be unrelated to IDD in the cervical spine. Individual differences and age may have a greater effect on IDD.

Degeneration refers to the state of the disc; it is not a diagnosis. It can have a number of possible causes [10]. Rather than considering it as an occurrence from a single process, it can be caused by multiple factors acting individually or collectively. Proper diagnosis requires clearly establishing the exact etiology. Degeneration occurring due to metabolic derangements is not difficult to diagnose because the clinical features of the primary condition will guide the diagnosis. Diagnostic challenges might occur in assessing whether the causes are genetic, increasing aging, or mechanical [11].

IDD can be caused by genetic factors that induce the production of abnormal matrix components, which distort the disc’s structure and function [9]. However, degeneration may not be wholly from genetic factors. Epidemiological studies revealed that genetic mutations may lead to higher of degeneration but may not be responsible for all cases nor for the variations across diverse ethnic populations. Population-level studies involving large, multiethnic groups are needed to clarify these concerns [16].

Alternatively, IDD is considered to originate from mechanical factors [11]. A rising body of evidence suggests the occurrence of vertebral endplate injury as a key mechanism. Injury may limit the disc’s nutrition indirectly or via a more direct process, triggering matrix degeneration. Changes in the biochemical profile associated with disc degeneration indicate connective tissue response to injury as they are idiopathic signs of aging. Evidence in favor of mechanical factors in the etiologic process is mainly from cadaveric investigations. Nevertheless, disc behavior is permanently altered in frozen animal or human cadaver specimens [17,18]. However, anyway, some evidence does support the mechanical factor hypothesis of disc degeneration, particularly evidence relating to torsional and compressive injuries. Compressive loading and torsion may result in endplate fracture and annular tea, respectively. These, sequentially, drive further biological changes.

Limited controversial evidence indicates a possible role of mechanical factors in the etiology of disc degeneration, which, when it involves the lower lumbar levels, is more common and more severe [16]. Discs just above a transitional vertebra usually have substantially higher degenerative changes compared to discs between a transitional vertebra and the sacrum [19]. Mechanical factors therefore were hypothesized to cause disc degeneration adjacent to a lumbar fusion [20,21].

Based on available evidence, degeneration of the intervertebral disc can be considered as an age-dependent, cell-mediated molecular degradation mechanism influenced by genetic mutations that are expedited mainly by nutritional and mechanical factors, and secondly by toxic or metabolic disorders [9,10,11,12]. The chemical interplay between these factors mediates degeneration. Degenerative impairment can modify the morphology of the disc (e.g., thickening of the vertebral endplate, cracks and fissures, and annular tear) and its biomechanical functioning. The end result may present as collapse of the intervertebral space and the formation of osteophytes [4,15].

Although genetics may explain the differences in the occurrence of disc degeneration, genetic mutations can also affect the anthropometry and the direct mechanical consequences of loading. Iatridis et al. postulated that this may be responsible for an unexplained variance of 25–50% [22]. Considering that intervertebral disc cells respond to mechanical loading based on the loading magnitude, frequency, and duration, they provide a framework arguing for a comprehensive biomechanical model of loads and the capacity of the disc to remodel determine what healthy and healthy loading is, and the response to loading [22].

A study examined the cervical spine and reviewed the incidence of prolapsed cervical intervertebral discs among professional drivers in Denmark over a decade; nearly all men in occupations involving professional driving had a statistically significant higher risk of hospitalization for a prolapsed cervical intervertebral disc. However, the risk in drivers who often performed heavy lifting was lower compared to those who rarely performed lifting. The authors hypothesized that the increased risk may relate to vibration and road shocks, twisting of the neck during acceleration and deceleration, and whiplash accidents, compared to heavy lifting, especially considering the current mechanization of loading and unloading [23]. Petersen et al. found no evidence of a positive association or an exposure-response effect of neck movements or neck positions on the risk of cervical disc herniation when using a job exposure matrix based on representative inclinometric measurements of the neck and register-based outcome measures [24].

Battie et al. evaluated male identical twins who were selected for discordance in suspected environmental risk factors [25]. A lifetime of occupational loading, including materials handling, positional loading, and bent and twisted postures, was persistently related to higher disc degeneration rate in univariate analysis, especially in the upper lumbar region. Interestingly, lower signal intensity was associated with higher physical loading during leisure-time, which supports the role of an environmental factors on disc signal. In multivariable analyses, type of labor accounted for only 7% of variability in the summary score of upper lumbar degeneration [26]. In the lower lumbar spine, heavy physical loading during leisure-time accounted for about 2% of the variability noted [25]. The researchers concluded that genetic mutations were more important; routine heavy physical loading at work and during leisure time explained only a limited overall variance in IDD between the twins. Interestingly, a longitudinal study of the same twins over a 5-year period showed that higher maximal lifting at work (but not occupational load) was related to higher reduction in lumbar disc height [26]. Williams et al., recently reported that occupation was not a significant determinant of back pain; however, this study was of female twins who had had limited exposure to heavy lifting activity [27].

The results of the present study show that occupation is independent of cervical spine degeneration. Age and individual differences underlie disc degeneration. The inclusion of asymptomatic cases may have contributed to the results of this study; different results might have been obtained by targeting symptomatic cases [14].

Our study has some limitations. First, we studied the Japanese population; thus, our results may not be generalizable. Second, we recruited a healthy volunteer, this might have led to selection bias in favor of healthy participants. Although all subjects were healthy, some had minor pathological and clinical problems, which might have affected the findings. Due to the excellent intra- and interobserver reproducibility of MRI measurements, measurements were performed only once by a single observer [28]. These objective measurements were carried out by an experienced radiology technician with an excellent knowledge of the human anatomy [8]. Our sample size was sufficiently large for evaluation. We compared individuals across age groups because this was a cross-sectional rather than a longitudinal investigation. We did not analyze the total amount of time the participants were working at their jobs. Since this study covers occupations and labor contents at the time of the survey, it is necessary to investigate the length of occupation and working hours in the next study. Cervical IDD was assessed using the modified Pfirrmann classification system, which has been used in previous studies and has proven high reliability [13]. However, future study should evaluate disc degeneration using a more comprehensive system, like the MRI grading scheme proposed by Matsumoto et al. [6]. Despite these limitations, this study was the largest of its kind, to our knowledge. It has strength in that all subjects were evaluated using the same imaging device. IDD patterns that result in symptoms can potentially be determined by comparing occupational and disc degeneration data from this study with those of symptomatic patients in further studies. As disc treatment develops, our results may serve as useful baseline data for planning clinical interventions.

## 5. Conclusions

In this cross-sectional survey of cervical spine MRI data among healthy adult volunteers, occupation and type of labor might have no effect on IDD in the cervical spine.

## Figures and Tables

**Figure 1 jcm-11-03301-f001:**
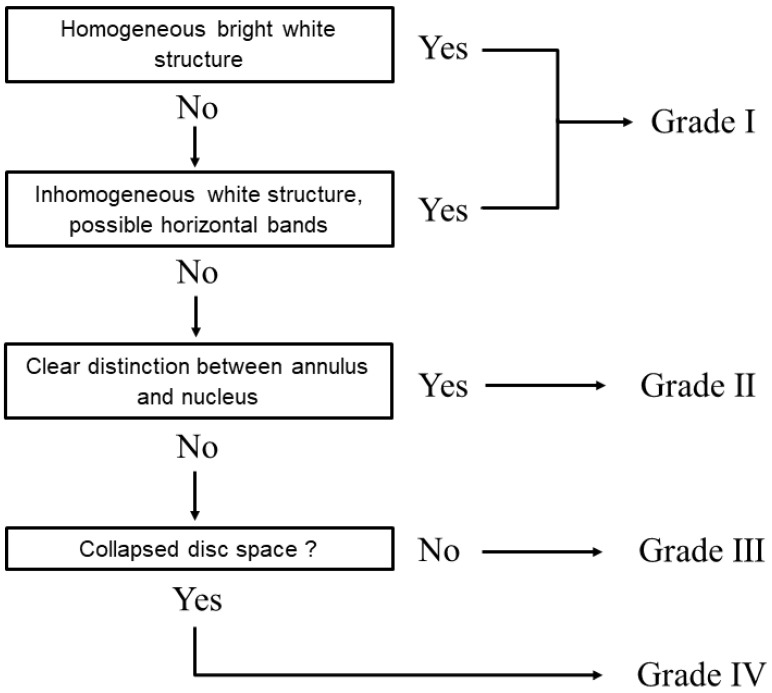
Algorithm for assessing the grade of cervical disc degeneration. The cervical disc degeneration grade is based on the modified Pfirrmann classification system.

**Figure 2 jcm-11-03301-f002:**
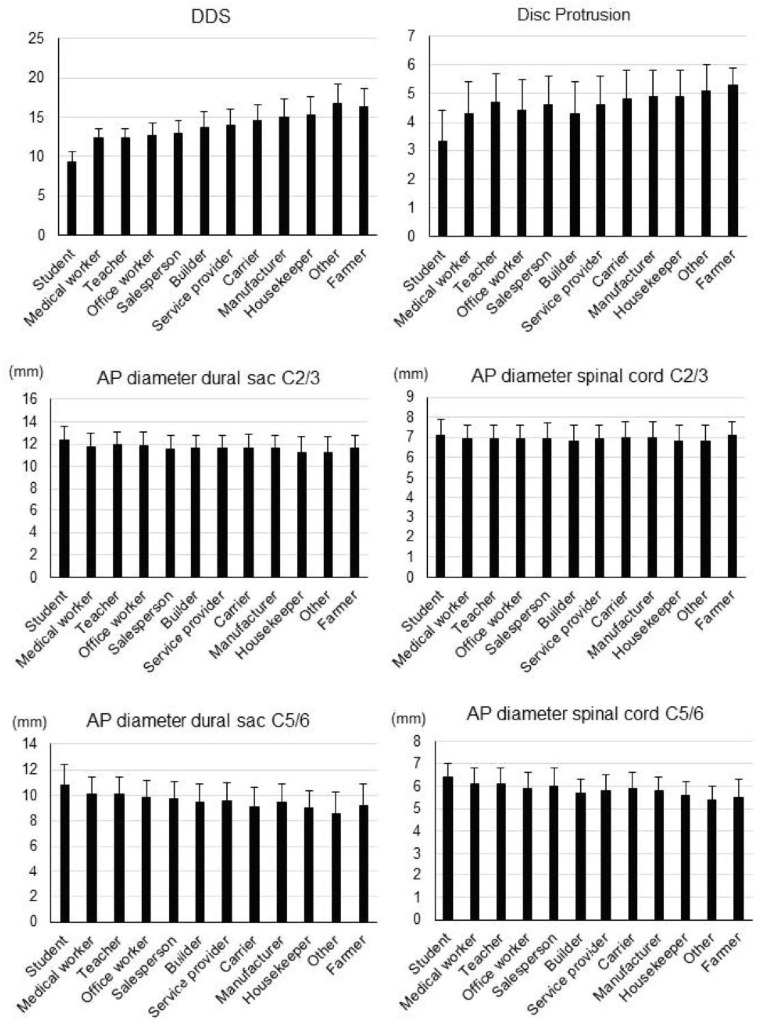
Disc degeneration data. Overall disc degeneration score (DDS), number of disc protrusions, and anteroposterior (AP) diameters of the dural sac and spinal cord at the C2/3 and C5/6 levels in each occupation group. Occupations are listed in ascending order of age.

**Figure 3 jcm-11-03301-f003:**
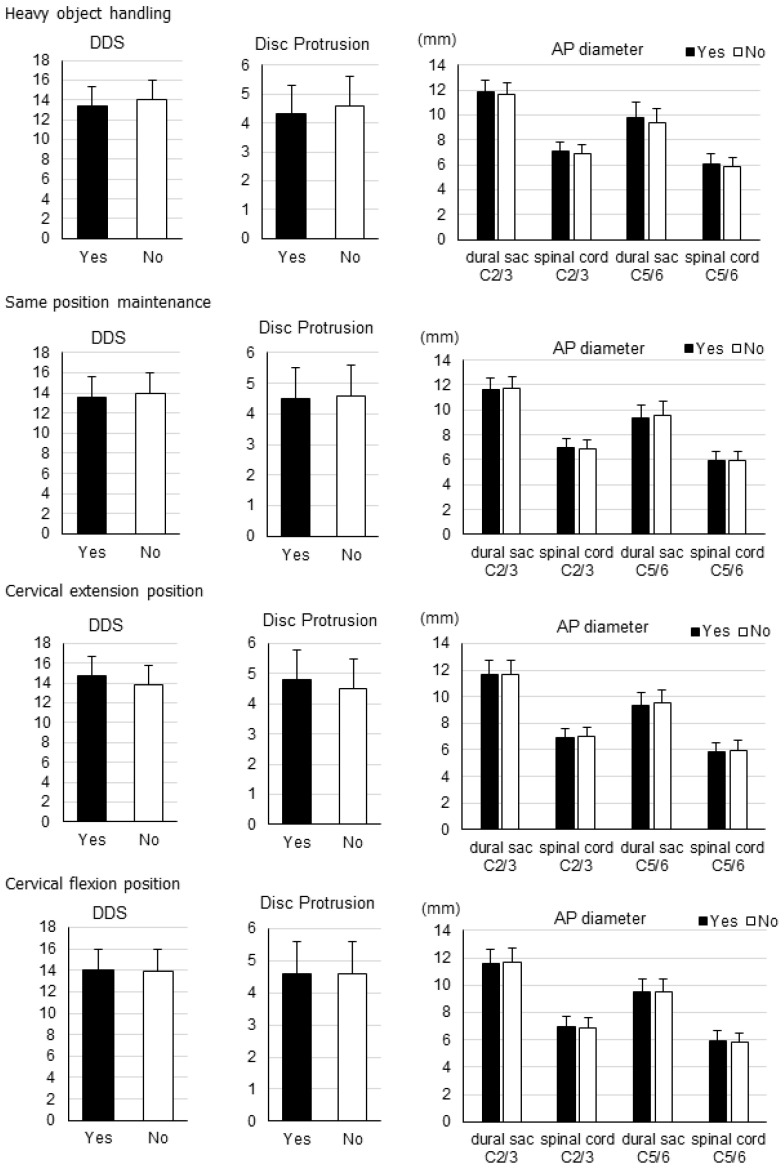
Disc degeneration data by labor type for males. Overall disc degeneration score (DDS), number of disc protrusions, and anteroposterior (AP) diameters of the dural sac and spinal cord at the C2/3 and C5/6 levels for each labor type in males.

**Figure 4 jcm-11-03301-f004:**
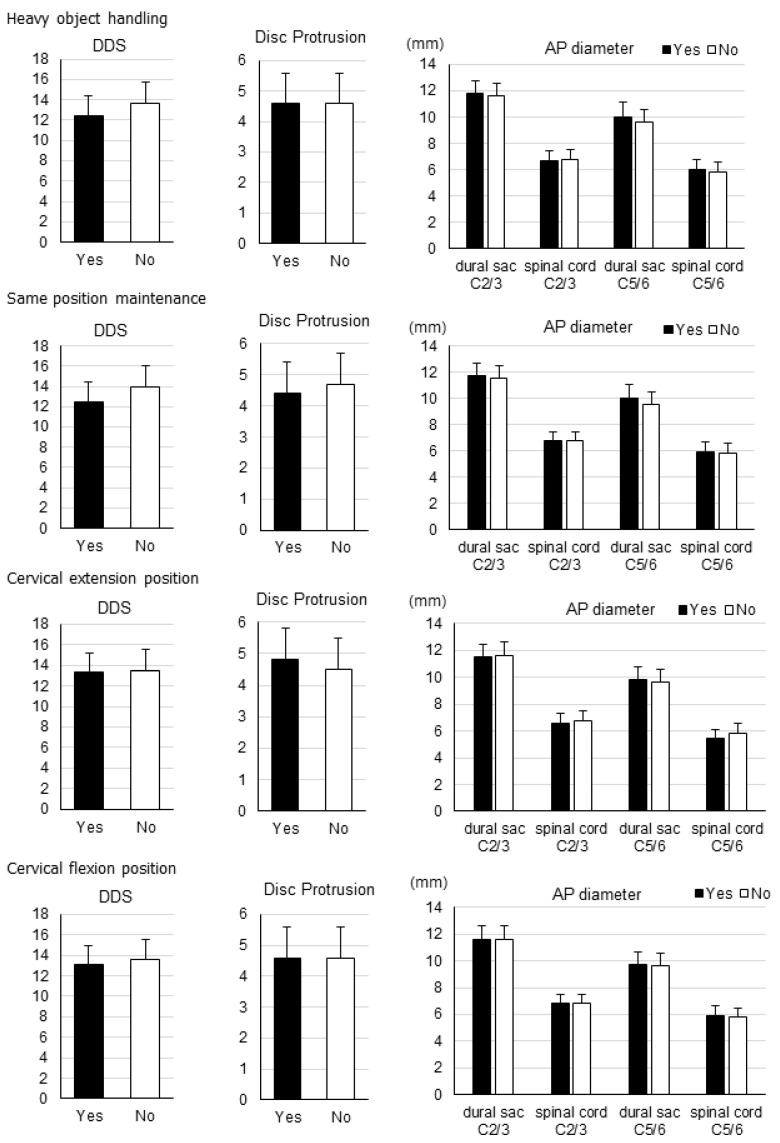
Disc degeneration data by labor type for females. Overall disc degeneration score (DDS), number of disc protrusions, and anteroposterior (AP) diameters of the dural sac and spinal cord at the C2/3 and C5/6 levels for each labor type in females.

**Table 1 jcm-11-03301-t001:** Demographics of asymptomatic subjects between decades or sexes.

Decade		Males	Females
20s	Number	101	100
	Age (years)	25.5 ± 2.6	25.5 ± 2.6
	Body height (cm)	172.4 ± 5.8	158.5 ± 5.5
	Body weight (kg)	65.6 ± 8.8	51.5 ± 6.0
	BMI (kg/m^2^)	22.1 ± 2.7	20.5 ± 2.3
30s	Number	104	99
	Age (years)	34.7 ± 2.7	34.8 ± 3.0
	Body height (cm)	172.5 ± 6.2	158.4 ± 6.0
	Body weight (kg)	70.3 ± 12.6	54.2 ± 8.1
	BMI (kg/m^2^)	23.6 ± 3.7	21.6 ± 3.1
40s	Number	100	100
	Age (years)	44.2 ± 3.0	44.2 ± 2.6
	Body height (cm)	171.0 ± 5.8	156.8 ± 5.2
	Body weight (kg)	69.4 ± 10.9	53.8 ± 8.4
	BMI (kg/m^2^)	23.7 ± 3.2	21.9 ± 3.3
50s	Number	99	103
	Age (years)	54.7 ± 2.6	54.6 ± 2.8
	Body height (cm)	168.6 ± 5.7	156.6 ± 5.8
	Body weight (kg)	67.6 ± 9.3	54.2 ± 8.1
	BMI (kg/m^2^)	23.8 ± 2.9	21.9 ± 3.7
60s	Number	101	103
	Age (years)	64.4 ± 2.6	64.4 ± 3.0
	Body height (cm)	165.5 ± 5.9	153.2 ± 6.0
	Body weight (kg)	64.0 ± 9.2	53.6 ± 7.8
	BMI (kg/m^2^)	23.3 ± 3.0	22.8 ± 2.8
70s	Number	101	100
	Age (years)	73.8 ± 2.6	73.1 ± 2.6
	Body height (cm)	162.4 ± 5.3	150.3 ± 5.2
	Body weight (kg)	61.5 ± 7.9	51.6 ± 7.7
	BMI (kg/m^2^)	23.3 ± 2.6	22.8 ± 3.0
Total	Number	606	605
	Age (years)	49.5 ± 16.9	49.6 ± 16.7
	Body height (cm)	168.7 ± 6.9	155.6 ± 6.3
	Body weight (kg)	66.4 ± 10.3	53.1 ± 7.8
	BMI (kg/m^2^)	23.3 ± 3.1	21.9 ± 3.2

Values given are mean ± standard deviation (SD) unless otherwise specified. BMI indicates body mass index.

**Table 2 jcm-11-03301-t002:** Occupation of 1211 asymptomatic subjects.

Occupation	Number	Age (Years)
Office worker	196	42.2 ± 13.3
Doctor, nurse, and medical coworker	196	37.7 ± 11.6
Housekeeper	193	60.9 ± 13.4
Service provider	101	50.7 ± 13.6
Builder	78	46.9 ± 14.3
Teacher	58	39.4 ± 13.8
Salesperson	57	42.6 ± 13.5
Manufacturer	54	56.6 ± 16.2
Student	16	22.8 ± 2.8
Carrier	15	53.4 ± 13.2
Farmer	3	71.3 ± 4.2
Unemployed person	124	50.2 ± 15.2
Other, Unknown	120	70.2 ± 6.2
Total	1211	49.5 ± 16.8

Values given are mean ± standard deviation (SD) unless otherwise specified.

**Table 3 jcm-11-03301-t003:** The detail of four labor types between males and females.

**Heavy Object Handling**				
	**Males**		**Females**	
	**Yes**	**No**	**Yes**	**No**
Number	116	428	84	420
Age (years)	46.3 ± 15.7	49.5 ± 16.9	42.8 ± 13.0	49.5 ± 16.0
**Same Position Maintenance**				
	**Males**		**Females**	
	**Yes**	**No**	**Yes**	**No**
Number	187	354	179	327
Age (years)	48.1 ± 14.4	50.0 ± 17.4	43.3 ± 13.0	51.8 ± 17.0
**Cervical Extension Position**				
	**Males**		**Females**	
	**Yes**	**No**	**Yes**	**No**
Number	33	498	13	480
Age (years)	51.9 ± 16.7	48.3 ± 16.8	45.4 ± 14.0	48.3 ± 16.0
**Cervical Flexion Position**				
	**Males**		**Females**	
	**Yes**	**No**	**Yes**	**No**
Number	142	395	145	362
Age (years)	49.8 ± 17.0	48.6 ± 16.0	45.4 ± 14.0	49.5 ± 17.0

Values given are mean ± standard deviation (SD) unless otherwise specified.

## Data Availability

The data of this study are available from the corresponding authors upon request.

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
