# Peer review of "Association between Occupation and Cervical Disc Degeneration in 1211 Asymptomatic Subjects"

_jcm, 2022, doi:10.3390/jcm11123301_

Round 1
Reviewer 1 Report
It is evaluated as an important study on the correlation between the degree of intervertebral disc degeneration according to occupational group and labor intensity. In the future, it is expected that detailed correlations according to the length of occupation and age will be studied.
Author Response
It is evaluated as an important study on the correlation between the degree of intervertebral disc degeneration according to occupational group and labor intensity. In the future, it is expected that detailed correlations according to the length of occupation and age will be studied.
→Thank you very much for your review. All authors totally agreed with your comment. We are interested the detailed correlations according to the length of occupation and age, next study will be performed in the future. Therefore, this content was described in limitation section of this manuscript. Once again, thank you for considering our manuscript.

Reviewer 2 Report
Machino et al have submitted an excellent manuscript describing the link (or absence thereof) bewteen cervical disk degeneration and occupation. The methodology and results are well described, and the conculsions are appropriate to the quality of data presented. A few recommendations to improve the manuscript:
- The authors describe 4 labor types: heavy object handling, same position maintenance, cervical extension position, and cervical flexion position. However it is unclear to the reader what occupations fall under each category. It is intuituvely understood that a builder or carrier would belong to the group "heavy object handling". But what occupations would fall in the group cervical extension position? Or cervical flexion position?
- The study sample is quite large and this adds validity to the results. However the manuscript should still clarify how the target sample size was calculated. What made the authors decide on recruiting 100 males and 100 females from each decade of age group? Was this based on some scientific method or simply convenience sampling? Either is fine but needs to be clearly stated.
- Some text is in larger font size compared to the rest. Please make entire manuscript font uniform.
- The methodology section contains a lot of references that appear to be needless. As an example, references 15-17 on line 95 are not needed. The statement "All images were transferred to a computer using imaging software" does not need a citation to support it. Same for references 18, 19 and 20 cited subsequently in the methodlogy.
Author Response
Machino et al have submitted an excellent manuscript describing the link (or absence thereof) bewteen cervical disk degeneration and occupation. The methodology and results are well described, and the conculsions are appropriate to the quality of data presented. A few recommendations to improve the manuscript:
→Thank you very much for your review. We would like to thank you for giving us the opportunity to revise our manuscript. We have carefully read all the remarks made by the referee and addressed all of your comments. All the passages that we have changed or added to the manuscript are highlighted on underline. Once again, thank you for considering our manuscript.
- The authors describe 4 labor types: heavy object handling, same position maintenance, cervical extension position, and cervical flexion position. However it is unclear to the reader what occupations fall under each category. It is intuituvely understood that a builder or carrier would belong to the group "heavy object handling". But what occupations would fall in the group cervical extension position? Or cervical flexion position?
→Thank you very much for your review. We agreed with following your excellent comments. As you pointed out, it is unclear what occupations fall under each category. Occupations and labors overlap, and this occupation cannot be strictly described as this labor alone. We think that the content requires further research in the future.
- The study sample is quite large and this adds validity to the results. However the manuscript should still clarify how the target sample size was calculated. What made the authors decide on recruiting 100 males and 100 females from each decade of age group? Was this based on some scientific method or simply convenience sampling? Either is fine but needs to be clearly stated.
→Thank you very much for your review. All authors totally agreed with your comment. Rather than the sample size calculated from statistical power, we recruited with this sample size because there is no study in the past that examined 100 males and females of each age group.
- Some text is in larger font size compared to the rest. Please make entire manuscript font uniform.
→Thank you very much for your review. We agreed with following your excellent comments. As you pointed out, we entired manuscript font uniform.
- The methodology section contains a lot of references that appear to be needless. As an example, references 15-17 on line 95 are not needed. The statement "All images were transferred to a computer using imaging software" does not need a citation to support it. Same for references 18, 19 and 20 cited subsequently in the methodlogy.
→Thank you very much for your review. We totally agreed with your appropriate opinion. According to your excellent suggestion, these references were deleted in this manuscript. Our study was improved to a better thing by your great suggestion. We appreciate heartily to your valuable comments.

Reviewer 3 Report
The article “Association between occupation and cervical disc degeneration in 1211 asymptomatic subjects” raises an interesting hypothesis if occupation and degeneration of the cervical disc are linked.
The authors had a large cohort of healthy study participants with impressive data on the first look.
Unfortunately, the study design is not well chosen. It is a cross-sectional design regarding the occupational status of the participants. The authors did not analyze the total amount of time the participants were working at their jobs (years working). Also, they did not check if they changed occupation recently.
Another problem is that the study has many students as participants.
Author Response
The article “Association between occupation and cervical disc degeneration in 1211 asymptomatic subjects” raises an interesting hypothesis if occupation and degeneration of the cervical disc are linked.
The authors had a large cohort of healthy study participants with impressive data on the first look. Unfortunately, the study design is not well chosen. It is a cross-sectional design regarding the occupational status of the participants. The authors did not analyze the total amount of time the participants were working at their jobs (years working). Also, they did not check if they changed occupation recently. Another problem is that the study has many students as participants.
→Thank you very much for your review. We would like to thank you for giving us the opportunity to revise our manuscript. We have carefully read all the remarks made by the referee and addressed all of your comments. All the passages that we have changed or added to the manuscript are highlighted on underline. Once again, thank you for considering our manuscript.
As you pointed out, we compared individuals between ages because this was a cross-sectional rather than a longitudinal investigation. We did not analyze the total amount of time the participants were working at their jobs (years working). This is a very important survey item and should be considered in the future. Since this study covers occupations and labor contents at the time of the survey, it is necessary to investigate the length of occupation and working hours. Therefore, these contents were described in limitation section of this manuscript. Because it was 16 out of 1211 subjects, students were included as a control for light work.
Our article was improved to a better thing by your excellent advice. We appreciate heartily to your excellent suggestion. Please consider this manuscript for publication. If you have any questions or requests, we will respond.

Round 2
Reviewer 3 Report
I acknowledge the changes made by the authors. Unfortunately, my comments were somehow not sufficiently addressed.
The authors still make conclusions that cannot be made based on their study design and results: “This study among healthy volunteers revealed that occupation had no effect on IDD. Surprisingly, individual differences and age had more impact on IDD than occupation.”
Another recent study demonstrated contradictory data, and the authors do not discuss this discrepancy. (cp. DOI: 10.1136/bmjopen-2021-053999).
All in all, I see little need for this study regarding its cross-sectional design, which is unable to prove or deny the hypothesis raised by the authors.
Author Response
I acknowledge the changes made by the authors. Unfortunately, my comments were somehow not sufficiently addressed. The authors still make conclusions that cannot be made based on their study design and results: “This study among healthy volunteers revealed that occupation had no effect on IDD. Surprisingly, individual differences and age had more impact on IDD than occupation.”
→Thank you very much for your review. We would like to thank you for giving us the opportunity to revise our manuscript. We have carefully read all the remarks and addressed all of your comments. All the passages that we have changed or added to the manuscript are highlighted on underline.
The authors totally agreed with your comment. As you pointed out, the conclusion may lead to confusion and misinterpretation for the readers. Then, to avoid misunderstanding for readers of this manuscript, the conclusion of our study was revised according to your appropriate opinion. We hope that the longitudinal study will find further conclusions in the future. Our article was improved to a better thing by your excellent advice. Once again, we appreciate heartily to your great help.
From
"This study among healthy volunteers revealed that occupation had no effect on IDD. Surprisingly, individual differences and age had more impact on IDD than occupation."
To
"In this cross-sectional survey of cervical spine MRI data among healthy adult volunteers, occupation and type of labor might have no effect on IDD in the cervical spine."
Another recent study demonstrated contradictory data, and the authors do not discuss this discrepancy. (cp. DOI: 10.1136/bmjopen-2021-053999).
→Thank you very much for your review. According to your appropriate suggestion, the content in another recent study was discussed substantially in this manuscript.
They found no evidence of a positive association or an exposure-response effect of neck movements or neck positions on the risk of cervical disc herniation (CDH) when using a job exposure matrix (JEM) based on representative inclinometric measurements of the neck and register-based outcome measures.
They used first diagnosis of CDH was retrieved from the Danish National Patient Register. They included the cases with the diagnosis of CDH from the registry and investigated the relationship with occupational neck exposures.
On the other hand, we evaluated the occupation and intervertebral disc degeneration (IDD) interplay using cervical spine magnetic resonance imaging (MRI) of relatively healthy subjects. We measured the anteroposterior diameter of the protrusion from the standard line to the posterior top of the protrusion, using a slice in which disc protrusion was most prominent in sagittal images.
Therefore, there are differences between our study and another recent study. Because we investigated the relationship between IDD on MRI images and occupation and type of labor in asymptomatic volunteers. Although JEM is an excellent measure of evaluation, the image had not been evaluated in another recent study. The disc bulging in our study is not also exactly CHD. Our study has strengths which details actual image evaluation.
As you pointed out, the manuscript was revised substantially, and the reference was added in the manuscript. Our article was improved to a better thing by your excellent advice. We would be extremely grateful if you would consider the original research article for publication in Journal of Clinical Medicine. Once again, we appreciate heartily to your great help.
The reference is as follows;
Petersen JA, Brauer C, Thygesen LC, Flachs EM, Lund CB, Thomsen JF. Prospective, population-based study of occupational movements and postures of the neck as risk factors for cervical disc herniation. BMJ Open. 2022 Feb 28;12(2):e053999. doi: 10.1136/bmjopen-2021-053999.
All in all, I see little need for this study regarding its cross-sectional design, which is unable to prove or deny the hypothesis raised by the authors.
→Thank you very much for your review. Magnetic resonance imaging (MRI) can be used as a noninvasive morphologic evaluation of the cervical spine. The signal intensity of the disc on MRI indicates the biochemical composition and likely pathologic changes. The decrease in signal intensity on T2-weighted MR image correlates with progressive degenerative changes of the intervertebral disc. Specifically, the brightness of the nucleus has previously been shown to be closely associated with the proteoglycan concentration of the disc. Although these degenerative changes in cervical disc MRI are often seen in healthy subjects, some could be pathological, but it is difficult to discriminate between normally aging discs and histologically degenerated discs which would lead to clinical symptoms. Therefore, it is essential to assess the frequency and degree of cervical disc degenerative changes seen on MRI in healthy subjects. Although the relationship between occupation and intervertebral disc degeneration (IDD) of the cervical spine is important, it remains unclear whether occupation affects IDD. Therefore, we conducted a large-scale study, across sexes and ages, to investigate the relationship between occupation and IDD.
We evaluated the MRI images of 1211 healthy individuals in detail. This study was the largest of its kind, to our knowledge. It has strength in that all subjects were evaluated using the same imaging device. IDD patterns that result in symptoms can potentially be determined by comparing occupational and disc degeneration data from this study with those of symptomatic patients in further studies. As disc treatment develops, our results may serve as useful baseline data for planning clinical interventions.
These contents were addressed in this manuscript. We believe strongly that the findings of this study will be of special interest to the readers of Journal of Clinical Medicine. We appreciate heartily to your valuable comments. If you have any questions or requests, we will respond.
The references are as follows;
- Tertti M, Paajanen H, Laato M, Aho H, Komu M, Kormano M. Disc degeneration in magnetic resonance imaging: a comparative biochemical, histologic, and radiologic study in cadaver spines. Spine (Phila Pa 1976). 1991 Jun;16(6):629–
- Pfirrmann CW, Metzdorf A, Zanetti M, Hodler J, Boos N. Magnetic resonance classifi cation of lumbar intervertebral disc degeneration. Spine (Phila Pa 1976). 2001 Sep 1;26(17):1873–1878.
- Boden SD, McCowin P, Davis D, Dina TS, Mark AS, Wiesel S. Abnormal magnetic-resonance scans of the cervical spine in asymptomatic subjects. A prospective investigation. J Bone Joint Surg Am. 1990 Sep;72(8):1178–1184.
